# The Use of Phosphate Washing Sludge to Recover by Composting the Leachate from the Controlled Landfill

Meriem Mobaligh [1], Abdelilah Meddich [2], Boujamaa Imziln [3] and Khalid Fares [1,*]

1   Laboratory of Pharmacology, Neurobiology, Anthropobiology and Environment, Faculty of Sciences Semlalia, Cadi Ayyad University, Marrakech 40000, Morocco; meriem.mobaligh.15@gmail.com
2   Laboratory of Agro-Foods, Biotechnologies and Valorisation of Plants Bioresources, Department of Biology, Faculty of Sciences Semlalia, Cadi Ayyad University, Marrakech 40000, Morocco; meddichabdelilah@yahoo.fr
3   Laboratory of Microbial Biotechnologies, Agroscience and Environment, Department of Biology, Faculty of Sciences Semlalia, Cadi Ayyad University, Marrakech 40000, Morocco; imziln@uca.ac.ma
*   Correspondence: fares@uca.ac.ma; Tel.: +212-661-232-285

**Abstract:** The percolation of rainwater and runoff water through household waste in the dumpsite generally leads to an overabundance of leachate in Moroccan landfills, which is a source of soil, surface water and groundwater contamination. In order to ecologically solve the problem posed by the leachate in the dump site, to safeguard the environment and to contribute to sustainable development, we have carried out this study which aims to study the possibility of composting leachate with green waste and phosphate washing sludge. Various combinations with five substrates (leachate, green waste, sugar lime sludge, phosphate washing sludge and olive mill wastewater) in different proportions were used to build five windrows. A 24 h contact between the phosphate sludge or sugar lime sludge and the leachate took place prior to the addition of the green waste for the construction of the different windrows. This contact time ensured the absorption of a significant portion of the leachate and the disappearance of bad odor. A significant reduction was obtained with streptococci and mesophilic flora after 24 h of contact. The monitoring of the physicochemical parameters throughout the composting process showed that the temperature of the different windrows followed a good pace presenting all composting phases. Moisture, pH, C/N ratio and the percentage of degradation of the organic matter conformed to the quality standards of the compost. The combinations of the alkaline treatment and the composting process allowed a significant hygienization of the leachate. The results of the humification parameters and the E4/E6 ratio suggest that the composts obtained with phosphate sludge were the most stable and mature and can be used in the agricultural field or green space.

**Keywords:** olive mill wastewater; leachate; compost; phosphate sludge; physicochemical and microbiological analyzes; sugar lime sludge



## 1. Introduction

In developing countries, solid waste management remains crucial because of demographic change, forced urbanization, and the improvement of living standards in each country [1]. In Morocco, the daily quantity of household waste generated is estimated at 18,000 tons, and this amount is increasing, with direct and indirect negative effects related to the nature and quantity of waste, its disposal, and treatment. This situation has prompted Morocco to opt for the technique of landfilling as a means of managing these huge quantities of waste characterized by their high organic matter content (65%) and high humidity (85%). However, the storage of these wastes at landfill sites produces harmful effluents (leachate), resulting from the combined action of rainwater and natural fermentation, known for their high loads of organic and mineral substances and thus pathogenic microorganisms, which creates a major environmental problem given the emissions of bad odors, soil and groundwater pollution [2].

Today in Morocco, in addition to the production of household waste, there are three other serious environmental problems such as that of the phosphate industry. It represents one of the main drivers of the country's economy. Morocco, with its large share of phosphate reserves (three-quarters of the world's phosphate reserves), is the world's leading exporter of phosphate derivatives, with an international market share of more than 30% [3]. This mining industry faces many major environmental challenges, resulting from the huge quantities (28.1 million tons per year) of phosphates washing sludge that Cherifian Phosphates Office (OCP) generates [4]. All these by-products are deposited together at the level of basins on the sites of the laundries, where they are stored. Moreover, the sugar industry generates an enormous amount of sugar lime sludge, by-products of the purification of beet juice, which increases each year with the increase in sugar beet and cane production. It is estimated that nearly 270,000 t of sugar lime sludge are produced each year [5]. This huge quantity of sugar lime sludge is always dumped outside the plants in wild lands without any valorization or treatment. Beyond their richness in calcium, phosphoric acid and magnesium assimilated by the plant are also considered perfect to activate agricultural soils and increase their pH [6], hence, the need for their exploitation. The olive industry is another kind of industry that constitutes a real polluting activity due to its production of huge quantities of olive mill wastewater. The discharge of these wastes into the environment without any prior treatment causes negative impacts due to their high organic loads, which are poorly biodegradable and highly toxic to plants, water, and soil microorganisms [7]. So far, these effluents have small economic value in Morocco.

Today, this situation of continuous accumulation of liquid and solid wastes (leachates, phosphate sludge, sugar lime sludge, and olive mill wastewater) encourages researchers to fit perfectly into the challenges of sustainable development by controlling and revalorizing these wastes on a large scale in order to protect the environment and fight against the resulting pollution in nature. From the point of view of the physicochemical characteristics of these substrates, composting seems to be the first effective solution for better ecological management, which will allow developing countries to engage with the stakes of the circular economy, which are at the same time an environmental, economic, and social issue.

In order to improve the performance of the composting process and to optimize the quality of final composts, we opted to carry out a comparison between the effect of using different concentrations of phosphate sludge and sugar lime sludge. Indeed, the physicochemical characteristics of phosphate sludge are almost identical to those of sugar lime sludge with regard to pH, organic matter, and total organic carbon. The main objective of this work was the valorization of phosphate sludge and sugar lime for the treatment by composting of leachates, and the elimination of their bad odors in order to obtain valuable composts that could be used in the agricultural valorization without negative effects on the soil and the crops. This study presents ecological and socio-economic benefits in the context of sustainable development related to the protection of the environment and natural resources through the recycling and valorization of large quantities of leachate.

## 2. Material and Methods

### 2.1. Composting Materials and Process Operation

Five types of waste including leachate (L), green waste (GW), phosphate sludge (PS), sugar lime sludge (SL), and olive mill wastewater (OMW) were used.

These wastes have various origins:

a    Olive mill wastewater is recovered from Bouchane oil mill, Morocco.
b    Green waste (grass) was collected from the garden of the Faculty of Science Semlalia Cadi Ayyad University of Marrakesh, Morocco. This type of green waste was chosen to ensure better absorption of the leachate.
c    The phosphate washing sludge was obtained from the Youssoufia region, Morocco.
d    Leachate was sampled from the ECOMED landfill in Marrakech, Morocco on 2 March 2020.
e    Sugar lime sludge was from the Doukkala sugar industry, Morocco.

The physicochemical properties of the initial substrates used in this work are presented in Table 1.

**Table 1.** Physicochemical properties of the initial substrate.

| Characteristics | L | GW | OMW | PS | SL |
|---|---|---|---|---|---|
| pH | 8.0 ± 0.04 | 7.9 ± 0.03 | 5.1 ± 0.01 | 7.9 ± 0.1 | 8.1 ± 0.05 |
| TKN (% DM) | 0.4 ± 0.1 | 2.8 ± 0.1 | 0.4 ± 0.1 | 0.04 ± 0.1 | 0.3 ± 0.1 |
| Organic matter (% DM) | nd | 81.2 ± 0.6 | nd | 10.7 ± 2.1 | 12.7 ± 0.2 |
| TOC (% DM) | nd | 45.1 ± 0.3 | nd | 5.9 ± 1.2 | 7.1 ± 0.1 |
| Humidity (%) | 95.7 ± 0.1 | 12.1 ± 0.6 | 90.5 ± 0.1 | 2.4 ± 0.1 | 4.7 ± 0.06 |
| $BOD_5$ (mg $O_2$/L) | 1400 ± 0.0 | nd | nd | Nd | nd |
| COD (mg $O_2$/L) | 25,750 ± 403.7 | nd | 190,550 ± 14,131.3 | Nd | nd |
| $BOD_5$/COD | 0.05 | nd | nd | Nd | nd |
| Ni | 0.07 ± 0.01 | 0.9 ± 0.3 | 0.01 ± 0.0 | 27.7 ± 1.1 | 0.8 ± 0.2 |
| Cu | 0.0 | 12.6 ± 0.1 | 0.0 | 32.4 ± 0.6 | 10.0 ± 0.2 |
| Pb | 0.01 ± 0.0 | 1.4 ± 0.1 | 0.01 ± 0.0 | 1.1 ± 0.01 | 0.7 ± 0.02 |
| Zn | 0.04 ± 0.0 | 55.8 ± 0.7 | 0.3 ± 0.01 | 302.2 ± 6.2 | 23.0 ± 0.6 |
| Cr | 0.07 ± 0.0 | 0.7 ± 0.04 | 0.0 | 51.4 ± 1.0 | 0.9 ± 0.02 |
| As | 0.5 ± 0.1 | 0.4 ± 0.1 | 0.0 | 19.6 ± 0.4 | 1.9 ± 0.02 |

L (leachate); GW (green waste); OMW (olive mill wastewater); PS (phosphate sludge); SL (sugar lime); TKN (total Kjeldahl nitrogen); TOC (total organic carbon); $BOD_5$ (biochemical oxygen demand); COD (chemical oxygen demand); Ni ( nickel); Cu (copper); Pb (lead); Zn (zinc); Cr (chromium); As (arsenic).

One day before composting is started (2 March 2020), we prepared five barrels with 80 L leachate in each barrel. This proportion was chosen in order to recover the largest possible quantity of final compost. A barrel control without any treatment was maintained and the other barrels were subjected to different treatments (Table 2) in order to test the effect of the alkaline sludge on the removal of fecal contamination indicators.

**Table 2.** Treatments carried out for each barrel.

| Barrels | Treatments | % of Sludge (*w/v*) |
|---|---|---|
| Barrel 1 | L | 0 |
| Barrel 2 | L + SL | 20 |
| Barrel 3 | L + PS | 20 |
| Barrel 4 | L + PS | 50 |
| Barrel 5 | L + PSW | 50 |

L (leachate); L + SL (leachate + sugar lime); L + PS (leachate + phosphate sludge); L + PSW (leachate + phosphate sludge + olive mill wastewater).

Subsequently, the barrels were carefully mixed and incubated 24 h at ambient temperature. This contact time is necessary for the reaction of chemical substances in the phosphate sludge and sugar lime sludge with the leachate. To maintain homogeneity of the mixture, the barrels were mixed five times during the contact time. After 24 h of contact between leachate/phosphate sludge and leachate/sugar lime sludge, we collected about 500 mL of sample from each treatment for physicochemical and microbiological analyses. Then, a sufficient quantity of green waste was added to each barrel in order to absorb the maximum of leachate and obtain a final C/N ratio between 25 and 28. The mixtures from each barrel were carefully mixed by turning to ensure good homogenization and the implementation of the windrows (1 m length, 0.7 m wide, and 0.5 m high) was performed on plastic sheeting to prevent leachate losses. The quantities (in kg) of the different wastes according to the windrows are shown in Table 3.

**Table 3.** Composition of the five windrows in kg.

| Windrows | GW (kg) | PS (kg) | L (L) | SL (kg) |
|----------|---------|---------|-------|---------|
| W1 | 33 | 0 | 80 | 0 |
| W2 | 33 | 0 | 80 | 16 |
| W3 | 33 | 16 | 80 | 0 |
| W4 | 33 | 40 | 80 | 0 |
| W5 | 33 | 40 | 80 | 0 |

GW (green waste); PS (phosphate sludge); L (leachate); SL (sugar lime).

### 2.2. Monitoring Composting

We conducted regular manual turning twice a week to ensure aeration of the windrows, as the aeration rate is considered one of the key factors affecting the composting process and the final quality of the compost; the windrows were watered whenever the moisture content dropped below 50%. It should be noted that windrow 5 was watered on day 7 and day 21 by olive mill wastewater (7 L in total).

The temperature was measured every day during the first week and once every 3 days for the rest. The value given corresponds to the mean of five measurements taken at different locations and varying depths of each windrow. At predetermined periods (start of composting, 7, 14, 21, 35, 49, 63, 77, 91, and 112 days), sampling was carried out uniformly throughout each windrow, at the surface and at different depths in order to obtain a homogeneous sample for physicochemical and microbiological analyses.

### 2.3. Physicochemical Analysis

The five windrows were characterized according to the main conventional physico-chemical parameters monitored during composting. The samples taken were subjected to various analyses, including the moisture content, which was determined on a 100 g sample after drying in an oven at 105 °C for 24 h. The pH was measured in a 1:10 ($w/v$) aqueous extract. The organic matter (OM) content was obtained by calcination at 650 °C for 6 h. The total organic carbon (TOC) was calculated according to Equation (1) [8]:

$$\text{COT (\%)} = \frac{OM\ (\%)}{1.8} \tag{1}$$

Total Kjeldahl nitrogen and assimilable phosphorus were determined according to the Kjeldahl method and the Olsen method, respectively. The heavy metals were determined by ICP-MS. All analyses were carried out in three replicates.

### 2.4. Determination of the Bacterial Population

Microbiological analyses were carried out on the leachate before any addition of sludge, on the leachate after 24 h contact with the sludge, and on the initial and final composting stages.

The samples collected were analyzed for microbial load indicators of fecal contamination and some pathogenic microorganisms. Fecal coliform and total coliform were determined on tergitol 7 and TTC agar. Bile esculin-azid agar was used for the enumeration of fecal streptococci. *Salmonella* spp. was identified using SS agar and confirmed on a triple sugar iron medium. Enumeration of the total mesophilic flora was performed on nutritious agar and diagnosis of *Pseudomonas aeruginosa* was performed using cetrimide agar as a selective medium.

### 2.5. Humification Process

In order to prove the stability and maturity of the five final composts, we calculated the humification indices (humification index (HI), humification rate (HR), and degree of humification (HD)), which are based on the quantification of the humic fraction compared to the fulvic fraction. For this purpose, we carried out the extraction of the total humic substances from the compost according to the method of [9]. The separation of humic (HA)

and fulvic (FA) acids was carried out by acidification with a sulphuric acid solution (6 N). The humidification rate, humidification degree, and humidification index are calculated according to Equations (2) [10], (3) [11], and (4) [12] respectively:

$$\text{Humification rate (HR\%)} = (\text{Carbon of total humic matter/Total organic carbon}) \times 100 \tag{2}$$

$$\text{Humification degree (HD\%)} = (\text{Carbon of total humic matter/Carbon of Humin}) \times 100 \tag{3}$$

$$\text{Humification index (HI)} = \text{Carbon of humic acid/Carbon of fulvic acid ratio} \tag{4}$$

*2.6. Statistical Analysis*

All the results are expressed as the mean and standard deviation. The physicochemical and microbiological results were analyzed by the ANOVA using XLStat Premium, version 2013.

**3. Results and Discussion**

*3.1. Moisture Content*

Generally, the moisture content is necessary for the bacterial activity and the degradation of organic matter during composting [13]. According to [14], the optimum moisture content is between 40% and 60%. The evolution of the moisture content of the different windrows during the composting process is shown in Figure 1. The results showed that windrows W1, W2, W3, W4, and W5 started with a moisture content of 73.2%, 67.5%, 64.2%, 58.2%, and 58.4%, respectively. The highest moisture content was recorded for W1, which does not contain any sludge, followed by W2 (20% of lime sludge). In fact, the results obtained in this study showed that phosphate sludge supplementation to the leachate decreased the moisture content as the amount added increased compared to the control. Thus, the phosphate sludge ensures better absorption of the leachate. The moisture content decreases in all windrows progressively to reach at the end of composting 53.3%, 45.5%, 42.2%, 40.7%, and 40.5%, respectively for W1, W2, W3, W4, and W5. This decrease could be attributed to the evaporation due to the increase in temperature generated by the microbial activity during composting [15]. Moreover, it is the consequence of the composting conditions, which we have proceeded to ensure the smooth running of the process, stirring, and aeration that leads to water loss in the form of steam. It should be noted that the difference between the windrow control and the windrow containing sugar lime sludge is significant ($p < 0.05$), but the difference between the windrow control and the windrows containing phosphate sludge is highly significant ($p < 0.01$).

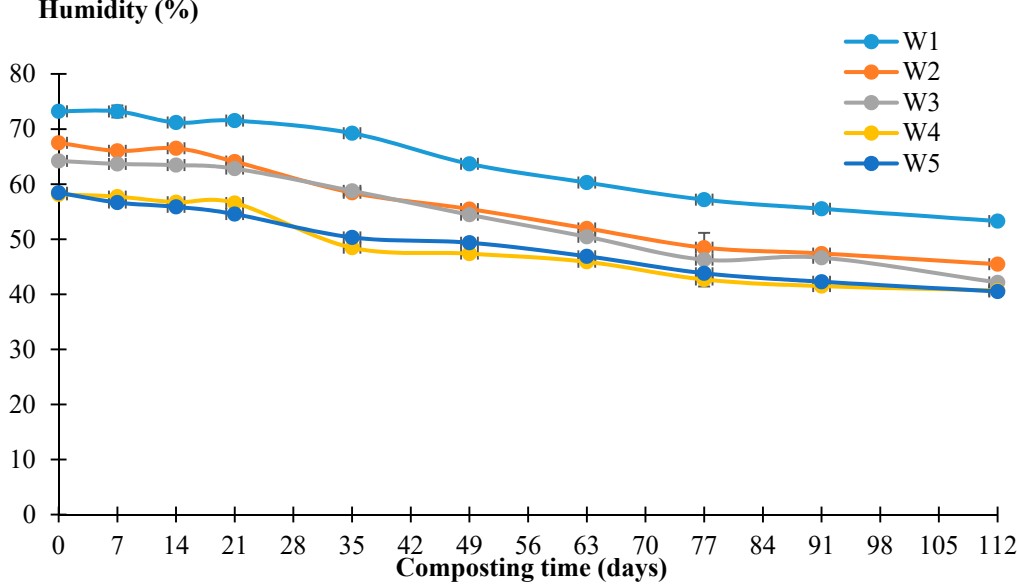

**Figure 1.** Variation of the humidity in the windrows during composting.

### 3.2. Temperature Evolution

According to [16], the success of composting depends on the temperature reached during the process. Indeed, the ability of the composting process to hygienize the compost from pathogens is associated with temperature [17]. The results in Figure 2 showed that the temperature curves for the five windrows had classic composting patterns, but no significant difference was detected between the control and the four different treatments. The thermophilic phase was established on the first day of composting and lasted almost 10 days, reaching maximum values of 54 °C for W1 and W3, followed by 53 °C for W4 and 51 °C for W2 and W5. This rapid increase observed during the initial phase is due to the degradation of simple compounds by microorganisms producing heat as a by-product [18,19]. After this phase, the temperature of the five windrows gradually decreased which may be in relation to the cold days as proved by the values of the ambient temperature. Then it increased again on the 63rd, 77th, and 84th days of composting. From 91 days, the temperature of the five windrows dropped to ambient temperature values, reflecting the maturity and stability of the composts. This indicates that simply metabolizable organic compounds have been decomposed and only compounds resistant to metabolization persist [20].

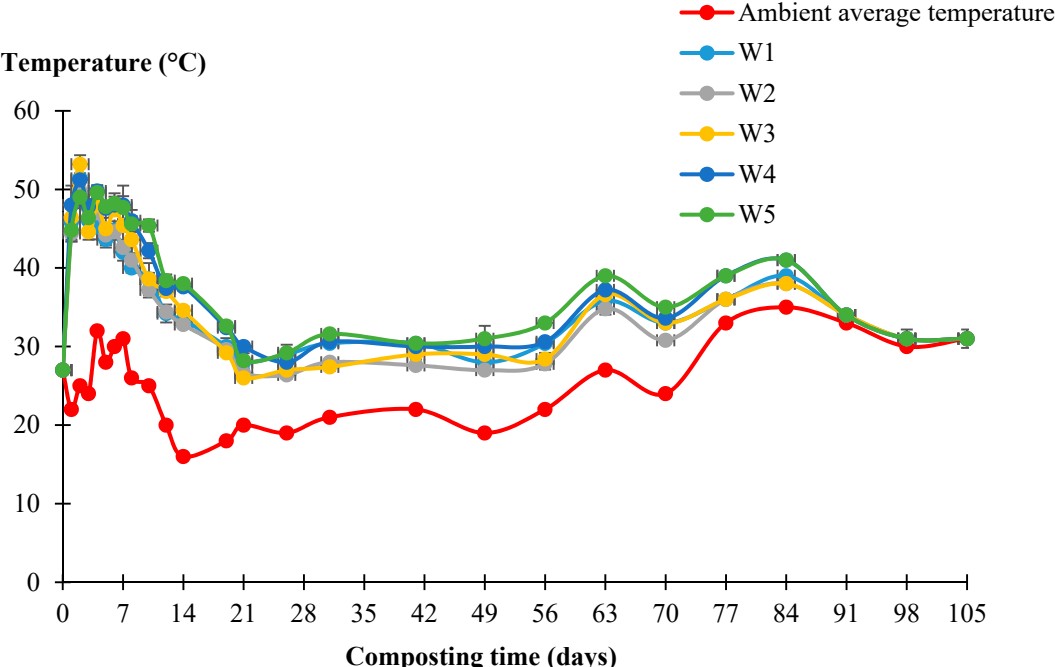

**Figure 2.** Temperature variation during the composting process.

According to [21], this last phase of composting (maturation) is characterized by the humification process, which consists of the polymerization of organic compounds into more stable compounds (humus).

### 3.3. Evolution of the pH

The evolution of pH during composting is shown in Figure 3. The change in pH of windrow 2 compared to the control was not significant, but the change in pH of windrows 3, 4, and 5 compared to the control was highly significant ($p < 0.01$). From the first week of composting, there was a slight decrease in the pH of the five windrows due to the accumulation of organic acids and the dissolution of $CO_2$ during the degradation of simple molecules [22]. The increase in pH for the five windrows in the following week could be explained by the ammonia production associated with the degradation of organic matter [23]. From day 21 to day 35, the pH of the five windrows decreases probably in relation to the volatilization of $NH_3^+$. Afterwards, a slight increase in pH was observed

in the five windrows due to the disappearance of easily degradable organic matter [24], and then the pH of the five windrows stabilized at relatively basic values: 8.6, 8.8, 8.8, 8.9, and 8.9 for W1, W2, W3, W4, and W5, respectively. Recommended pH values for mature compost are normally between 7 and 9 [25]. Throughout the composting process, and even with the same concentration of phosphate sludge, windrow 5 showed a lower pH than windrow 4, which could be logically associated with watering by the olive mill wastewater, which has a pH = 5.1 (Table 1).

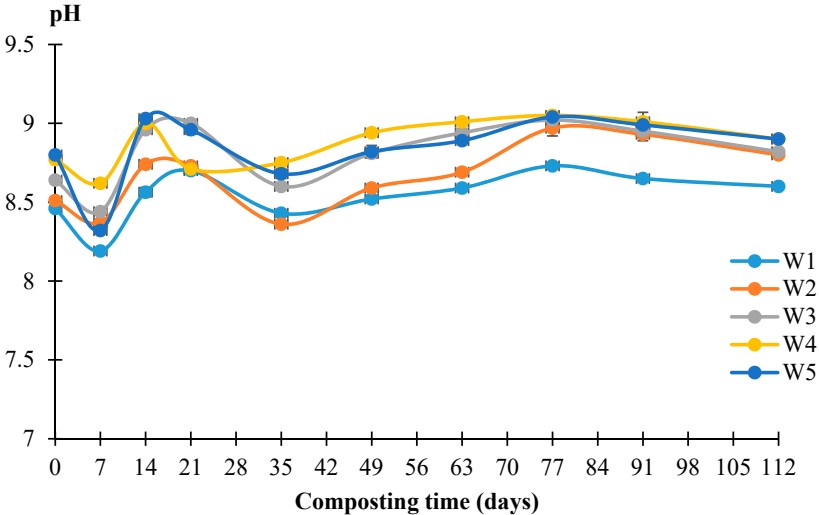

**Figure 3.** Evolution of the pH in the different windrows during the composting process.

### 3.4. C/N Ratio Evolution and Organic Matter Degradation

The C/N ratio is one of the factors used to control the composting process [26]. The C/N ratio decreased gradually and followed an almost similar trend in the five windrows (Figure 4). This is probably due to the loss of carbon in the form of carbon dioxide due to the degradation of organic matter and the increase in total nitrogen content due to organic matter mineralization of the initial substrates by the microorganisms [27]. At the end of composting, the C/N ratio of the final composts was 11.8, 13.0, 13.7, 12.2, and 12.7 for windrows 1, 2, 3, 4, 5, respectively, which proves that the final composts were mature and stable, as they were within the range recommended by [28].

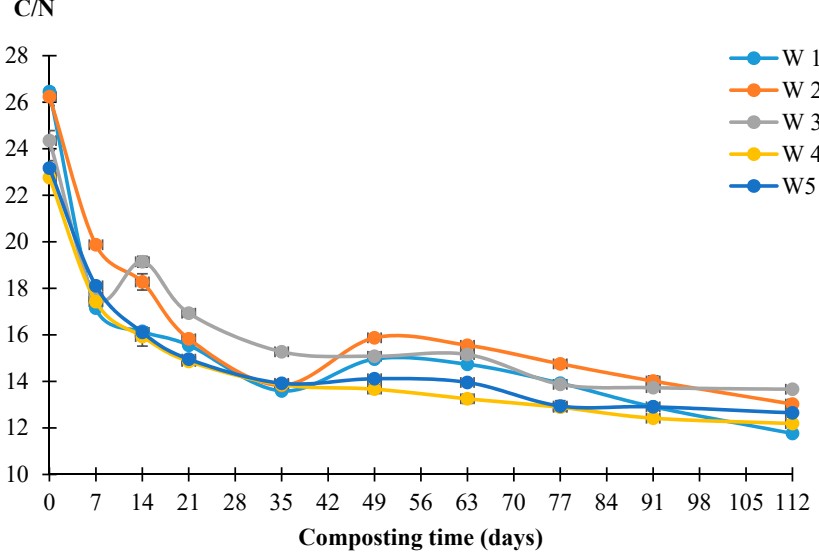

**Figure 4.** Evolution of the C/N ratio during composting.

The organic matter content of the final composts was within the range recommended by [27]. It was in the order of 53.0%, 38.0%, 31.9%, 24.6%, 24.6%, and 25.0% for W1, W2, W3, W4, and W5, respectively. The maximum organic matter degradation (Figure 5) was observed for W4 (34.1%), followed by W3 (33.9%), W5 (32.5%), and W2 (30.1%), while for W1, which did not contain any sludge, the organic matter degradation was only 15.1%. These results showed that the addition of phosphate sludge or sugar lime sludge had a significant impact on the degradation of organic matter. The percentage of organic matter degradation was not similar in the two windrows W4 and W5 that underwent the same treatment (50% PS) due to the fact that windrow 5 was watered by olive mill wastewater on day 7 by 5 L and day 21 by 2 L. This result showed that the 7 L watering by the olive mill wastewater contributed to a slight increase in organic matter in windrow 5 of about 1.9% DM compared to windrow 4.

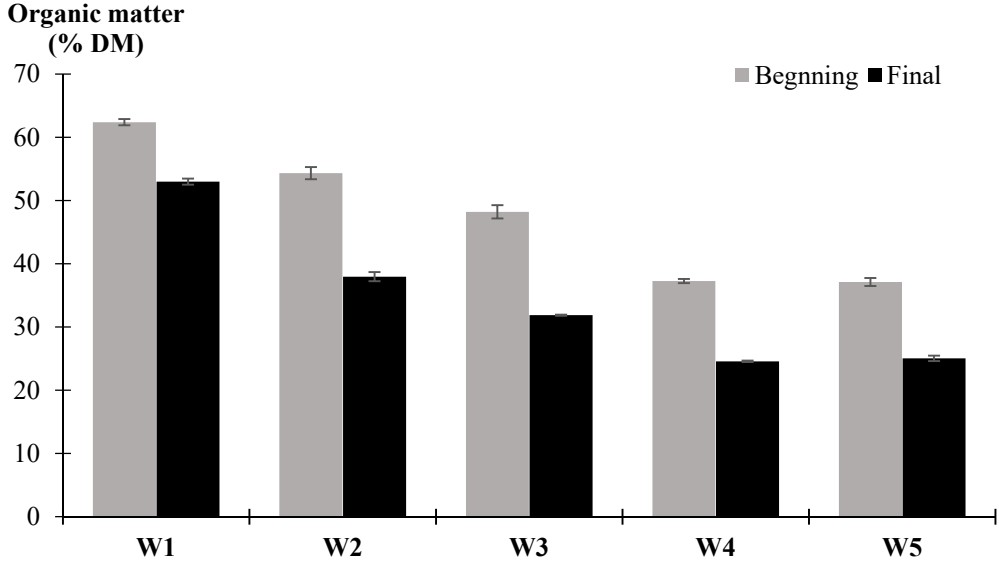

**Figure 5.** Organic matter contents in the different windrows.

### 3.5. Monitoring of Microbiological Parameters

3.5.1. Microbiological Characteristics of the Raw Leachate

Microbiological analyses of the raw leachate (Table 4) used in this study showed a total absence of *Salmonella* spp. and *Pseudomonas aeruginosa*. The absence of these pathogenic microorganisms could be explained, among other things, by the alkaline nature of the leachate, which is unfavorable for the development of these bacteria [29]. Total and fecal coliforms are frequent in the environment. However, their absence in the leachate sample may be due to the absence of favorable conditions for their development. Indeed, results relating to the evolution of fecal coliforms suggest that these microorganisms are seasonally dependent: a decrease in the bacterial load of fecal coliforms during the cold season has been recorded [30].

**Table 4.** Results of the microbiological analyses on raw leachate from the landfill of Marrakech.

| Microorganismes | CFU/mL |
|---|---|
| Fecal streptococci | 9050 |
| Fecal coliforms | 0 |
| Total coliforms | 0 |
| Total mesophilic flora | 163,500,000 |
| *Pseudomonas aeruginosa* | 0 |
| *Salmonella* spp. | 0 |

An abundance of fecal streptococci was detected in the leachate sample. Indeed, fecal streptococci are very good indicators of fecal contamination and are more resistant to environmental factors than coliforms [31]. A high concentration of mesophilic flora was also observed. In general, the total mesophilic flora provides information on the indigenous microflora brought by pollution; it is used as an indicator of overall pollution [30].

3.5.2. The impact of Adding Phosphate Sludge on the Content of Microorganisms in the Leachate

Before starting the composting process, we proceeded to carry out a direct contact of 24 h between the sludge and the leachate, in order to evaluate the effect of the different treatments on the bacterial load of the raw leachate.

The addition of phosphate sludge and sugar lime sludge allowed a significant reduction in fecal streptococci and total mesophilic flora (Figures 6 and 7); almost similar trends were observed for the two treatments phosphate sludge 20% and sugar lime sludge 20%. With 20% sugar lime sludge, a 53.7% reduction in fecal streptococci was observed, while with 20% of the phosphate sludge, a 51.9% destruction was observed after 24 h of contact. When the phosphate sludge addition was increased to 50%, 56.4% to 57.8% destruction of fecal streptococci was observed.

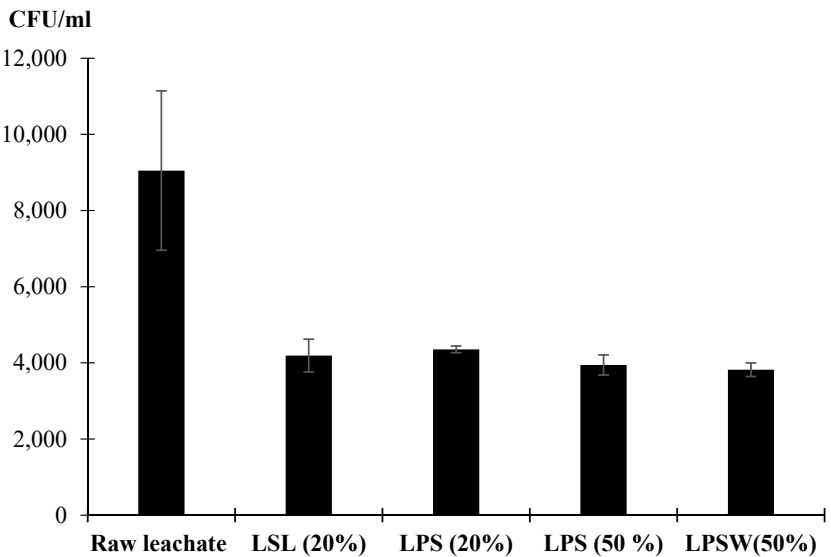

**Figure 6.** Evolution of the fecal streptococci in the different treatments after 24 h of contact.

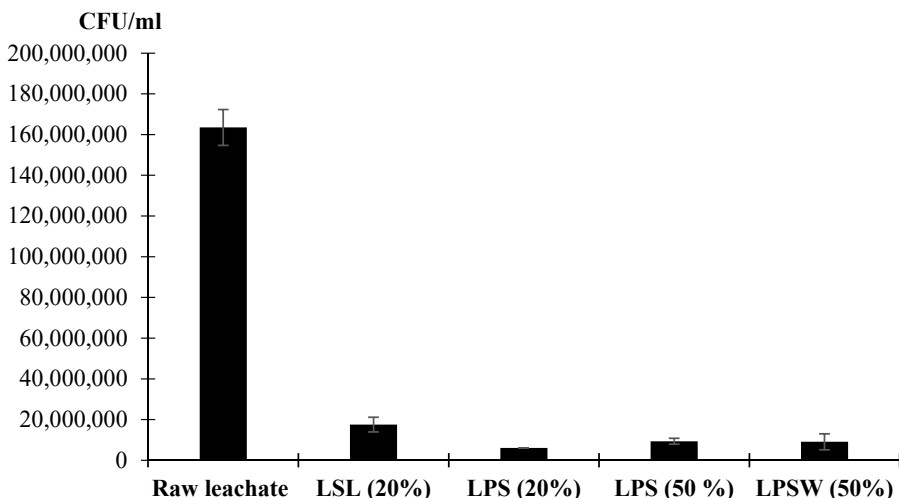

**Figure 7.** Evolution of the total mesophilic flora in the different treatments after 24 h of contact.

The concentration of the total mesophilic aerobic flora was considerably destroyed after 24 h of contact for all four treatments: 89.3%, 96.3%, 98.6%, and 98.4% for W-SL (20%), L-PS (20%), L-PS (50%), and L-PSW (50%), respectively. The percentage of reduction in total mesophilic flora was higher than that of fecal streptococci, indicating its sensitivity to lime.

The treatments have successfully reduced the pathogenic agents. These results confirm those found by [5], which demonstrated the effectiveness of alkaline treatments in pathogen reduction. Although pathogen reduction was satisfactory, a complete composting cycle was necessary for better hygienization of the final products.

### 3.5.3. The Effect of Composting on the Hygienization

To evolve the effect of composting as a hygienization process, we conducted counts of the microbial community considered at the beginning and the end of the composting process. Although there was a significant drop in the total mesophilic flora due to the addition of phosphate sludge and sugar lime sludge, the addition of green waste was satisfactory in terms of increasing the total mesophilic flora (as well as fecal streptococci) required to start composting Figure 8. At the beginning of composting, the total mesophilic flora was predominant: $1 \times 10^9$ CFU/g for W1, $8.45 \times 10^8$ CFU/g for W2, $7.8 \times 10^8$ CFU/g for W3, $6.5 \times 10^8$ CFU/g for W4, and $5.65 \times 10^8$ CFU/g for W5. Towards the end of composting, this flora remained abundant in the final composts: $1.6 \times 10^8$ CFU/g, $1.32 \times 10^8$ CFU/g, $8.73 \times 10^7$ CFU/g, $1.03 \times 10^8$ CFU/g, and $1.07 \times 10^8$ CFU/g for W1, W2, W3, W4, and W5, respectively. This could be explained by the environmental conditions that favor the reinstallation of the new mesophilic flora, which is essential for obtaining a mature product [32]. Although there was a reduction in fecal streptococci in the control (90%), a high level of fecal streptococci ($2.83 \times 10^3$ CFU/g compost) was still observed even after 112 days of composting. However, treatment with sugar lime sludge resulted in a greater reduction than the control (97.5%). No fecal streptococci were detectable after 112 days in the three composts treated with phosphate sludge. This reduction could be attributed to the alkaline treatment and the hygienization process known in the composting operation. The study of [33] considered the increase in temperature and the increase in pH that leads to the release of ammonia as the main factors involved in the reduction of pathogens from biosolids during composting. We can conclude that the combination of the alkaline treatment and the composting process stabilized the final products to meet those recommended in [28].

### 3.6. Humification Process during Composting

According to [34], the monitoring of humification parameters during composting could be a representative index of the evolution of compost maturity and stability, since compost quality is confirmed by the quantity of stable humus formed after the biodegradation of organic matter [35]. The results reported in Table 5 showed that at the end of composting (112 days), all composts containing phosphate sludge had their degree of humification and their humification rate above 70% and 35%, respectively, which are considered standards, indicating a good humification [36]. Similarly, the CAH/CFA ratio was 1.9 for compost 3, 3.5 for compost 4, and 3.0 for compost 5. These values were higher than the 1.6 value estimated by [37] for stabilized organic matter.

In contrast, the E4/E6 ratio for the composts treated with phosphate sludge and sugar lime sludge is less than five, indicating that the composts are mature and its humus particles are complex [38]. The E4/E6 ratio of the control is higher than five, which proves that the degradation of organic matter in the control (without sludge) was less advanced and that the humic substances formed are not stable. According to [39], an E4/E6 ratio higher than five shows the presence of fulvic and the formation of small molecules.

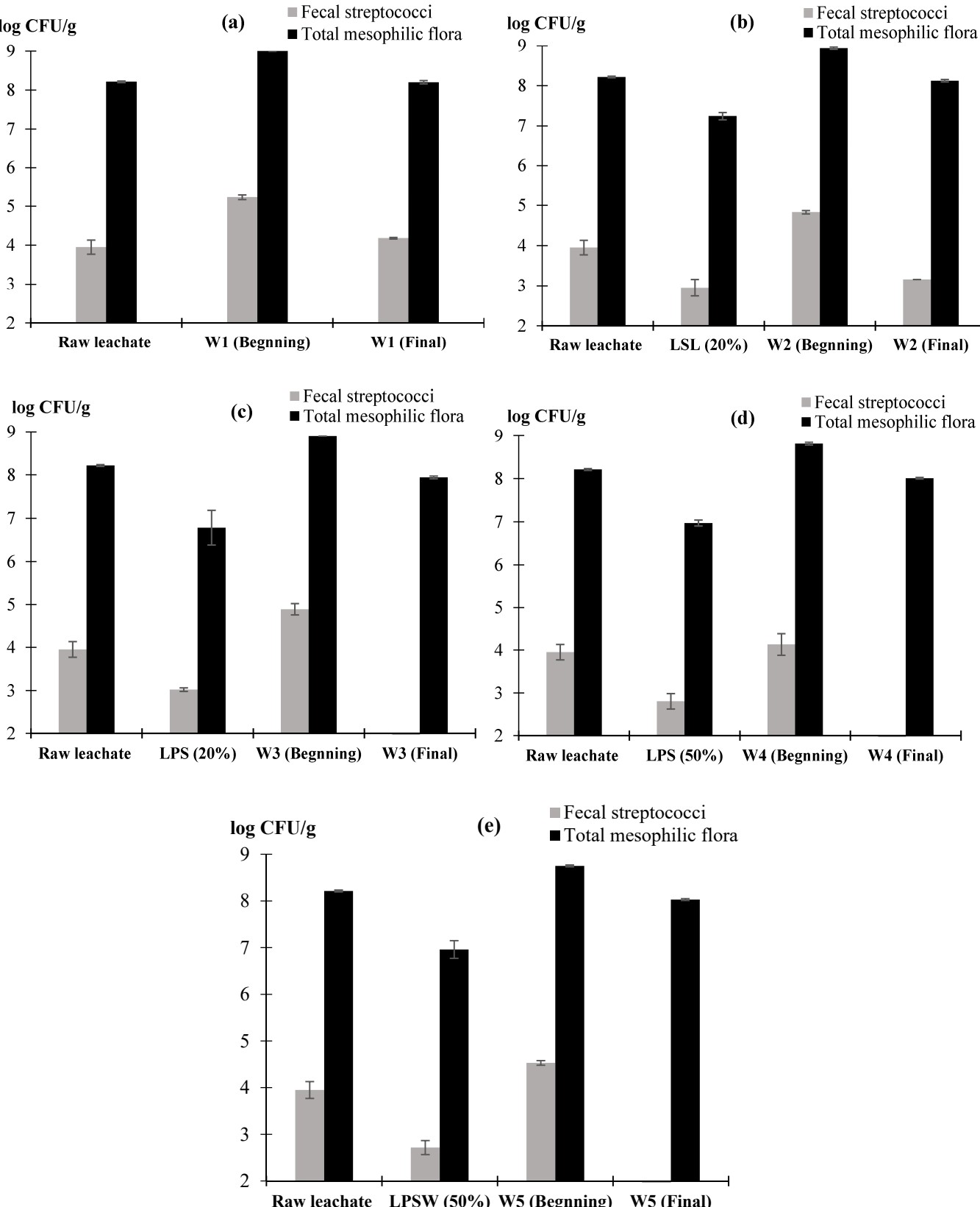

**Figure 8.** Results of microbiological analyses on the windrow 1 (**a**), windrow 2 (**b**), windrow 3 (**c**), windrow 4 (**d**), and windrow 5 (**e**) at the beginning and after 112 days of composting.

**Table 5.** Humification indices of the five final composts.

| Composts | C1 | C2 | C3 | C4 | C5 |
|---|---|---|---|---|---|
| Humification index (IH) | 0.9 ± 0.1 | 1.3 ± 0.2 | 1.9 ± 0.2 | 3.5 ± 0.6 | 3.0 ± 0.2 |
| Humification rate (HR%) | 27.0 ± 0.7 | 41.7 ± 0.6 | 53.9 ± 0.6 | 78.1 ± 2.4 | 76.6 ± 0.2 |
| Humification degree (HD%) | 43.6 ± 0.0 | 59.2 ± 5.5 | 78.4 ± 2.4 | 95.7 ± 0.3 | 97.8 ± 2.7 |
| E4/E6 | 11.8 ± 0.6 | 3.5 ± 0.3 | 2.7 ± 0.1 | 3.5 ± 0.2 | 3.7 ± 0.1 |

According to these results, the addition of phosphate sludge has a significant effect on the humification index, the humification rate, the degree of humification, and the E4/E6 ratio compared to the control and even compared to the compost treated with sugar lime sludge. Therefore, the addition of phosphate sludge allowed obtaining a more stable and mature compost compatible with the use in agricultural fields or green spaces.

### 3.7. Effect of the Composting Process on Available Phosphorus Content

As shown in Figure 9, a significant increase in available phosphorus content from the beginning to the end of the composting process was recorded in the five composts. This increase was in the order of 34.1%, 34.7%, 31.4%, 53.5%, and 40.7% for C1, C2, C3, C4, and C5, respectively. These results could be explained by the activity of phosphorus solubilizing microorganisms during the composting process to release more assimilable phosphorus than it was in the initial substrate. The difference in average available phosphorus content at the beginning and final stages of the process could also be attributed to the initial organic matter composition of each windrow (Figure 5), which directly influences the available phosphorus content in each compost. According to [40], the form and rate of phosphorus change during the composting process is influenced by the type of waste used to prepare the compost and the degradation of organic matter by microorganisms.

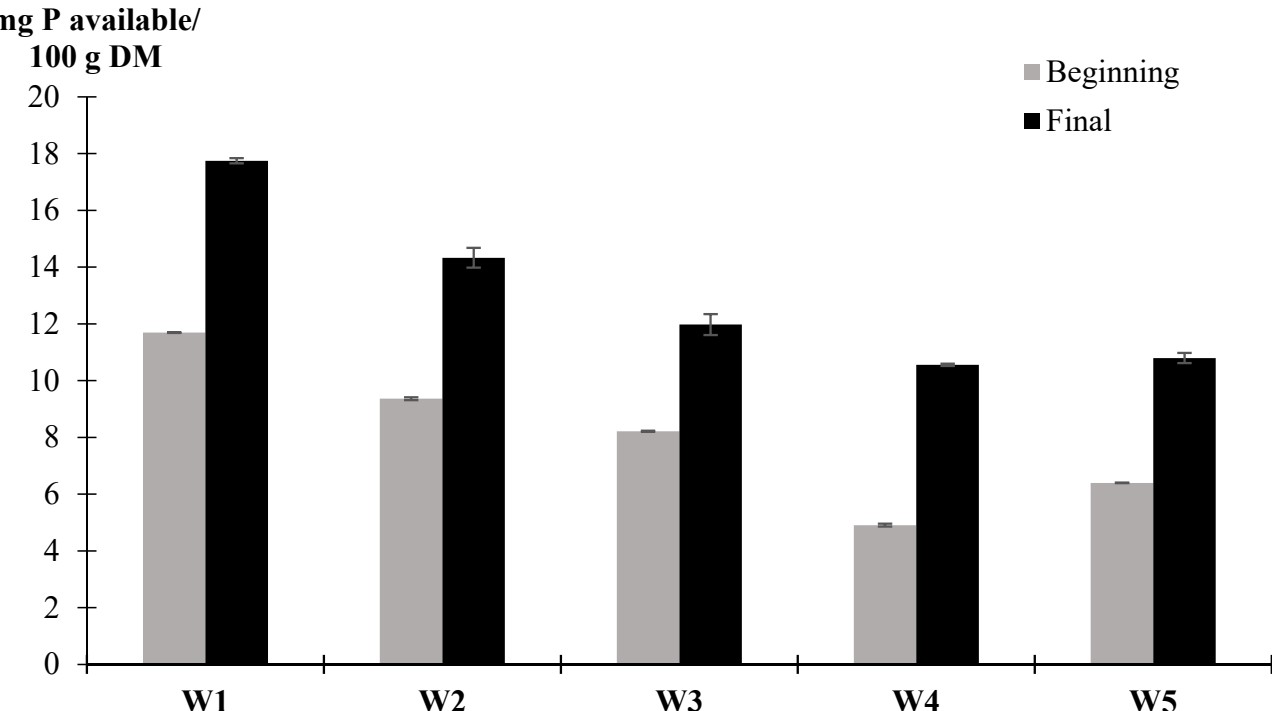

**Figure 9.** Available phosphorus content in the five windrows at the initial and final phases of the composting process.

These results require a complete and detailed speciation of phosphorus and its dynamics during composting to understand its evolution and bioavailability.

### 3.8. Visual Quality of the Final Composts

The results of Table 6 show the effect of the addition of phosphate sludge and sugar lime sludge on the final quality of the composts. For the control (C1), which was not treated, the color was the darkest compared to the other composts. The two composts obtained with 20% of sugar lime sludge (C2) and phosphate sludge (C3) had a similar color (brown). However, by increasing the concentration of phosphate sludge, the final color of the compost 4 can be less brown. Watering windrow 5 with the olive mill wastewater modified the final color of the compost 5. Compared to the control (C1), the treatment of 50% of phosphate sludge ensured a finer texture similar to that of soil characterizing a good degradation of the material during the composting cycle, followed by the two other treatments of 20% of phosphate sludge and sugar lime sludge. The unpleasant odor of the leachate has completely disappeared. In addition, there was a complete absence of impurities (plastic, glass, etc.) in the five composts as stipulated by [28].

**Table 6.** Macroscopic observations on the final composts.

| | Compost 1 (C1) | Compost 2 (C2) | Compost 3 (C3) | Compost 4 (C4) | Compost 5 (C5) |
|---|---|---|---|---|---|
| Texture | Fine to coarse | Fine to coarse | Fine to coarse | Fine to coarse to mostly fine | |
| Color | Dark brown | Brown | Brown | Light brown | Brown |
| Bad odor | Absent | Absent | Absent | Absent | Absent |

### 3.9. Heavy Metals Concentration

The concentration of heavy metals is considered among the main factors that affect the quality of compost and limit its use and marketing [41]. During the composting the concentration of heavy metals may decrease or increase [42], because the water evaporation which occurs during composting and the solubilization of heavy metals [43] influenced by the physicochemical changes (pH, CEC, $NH_4^+$, $NO_3^-$).

Six heavy metals: nickel, chromium, copper, zinc, lead, and arsenic were analyzed at the beginning and at the end of the process. The results (Table 7) revealed that the heavy metal contents of the windrows varied from one treatment to another. The increase observed in Cu, Zn, Ni, Cr, and As contents towards the end of the process could be due to weight loss due to the degradation of organic matter and loss through respiration during the composting processes [43,44]. A similar result for the increase in Zn, Cr, and Cu has been shown by [41]. On the contrary, Pb concentration decreased towards the end of composting, with the exception of windrows 2 and 3 where an increase was observed. This result is in agreement with that of [45] who showed that when $SO_4^{2-}$ and $PO_4^{3-}$ increase the lead content decreases and also the results obtained by [46] and [41] who found that the lead content increases during composting. The concentrations of heavy metals analyzed in the five final windrows (produced composts) were very low compared to the range concentrations defined by the [28]. Therefore, the composts produced do not contain heavy metals in proportions probable to present a risk for soil, plant, or groundwater contamination.

**Table 7.** Heavy metals concentration in windrows compared to the French Norm (NF U44-051).

| Hevay Metals | Windrows | Begnning | Final | NF U44-051 Range |
|---|---|---|---|---|
| **As** | W1 | 3.8 ± 0.02 | 8.9 ± 0.02 | 18 mg/kg DM |
| | W2 | 3.4 ± 0.04 | 5.3 ± 0.01 | |
| | W3 | 6.5 ± 0.04 | 7.7 ± 0.1 | |
| | W4 | 5.2 ± 0.1 | 7.3 ± 0.04 | |
| | W5 | 4.0 ± 0.03 | 6.6 ± 0.3 | |
| **Cr** | W1 | 1.5 ± 0.2 | 5.4 ± 1.9 | 120 mg/kg DM |
| | W2 | 1.6 ± 0.3 | 2.2 ± 0.1 | |
| | W3 | 13.1 ± 1.2 | 10.7 ± 1.8 | |
| | W4 | 13.8 ± 2.0 | 14.9 ± 1.3 | |
| | W5 | 10.4 ± 0.7 | 13.8 ± 0.8 | |
| **Zn** | W1 | 48.6 ± 0.6 | 95.9 ± 5.5 | 600 mg/kg DM |
| | W2 | 39.8 ± 1.0 | 50.5 ± 1.6 | |
| | W3 | 103.5 ± 1.0 | 125.6 ± 1.6 | |
| | W4 | 98.3 ± 3.4 | 132.7 ± 2.4 | |
| | W5 | 79.8 ± 1.2 | 121.6 ± 10.5 | |
| **Cu** | W1 | 12.7 ± 0.1 | 26.2 ± 1.0 | 300 mg/kg DM |
| | W2 | 8.5 ± 0.2 | 12.9 ± 0.1 | |
| | W3 | 18.5 ± 0.2 | 17.7 ± 0.1 | |
| | W4 | 12.9 ± 0.6 | 16.8 ± 0.3 | |
| | W5 | 10.2 ± 0.1 | 16.4 ± 0.6 | |
| **pb** | W1 | 2.3 ± 0.2 | 1.4 ± 0.1 | 180 mg/kg DM |
| | W2 | 0.5 ± 0.04 | 0.9 ± 0.04 | |
| | W3 | 0.4 ± 0.05 | 0.7 ± 0.03 | |
| | W4 | 1.9 ± 0.1 | 0.6 ± 0.02 | |
| | W5 | 2.4 ± 0.1 | 0.8 ± 0.04 | |
| **Ni** | W1 | 1.8 ± 0.1 | 6.3 ± 0.8 | 60 mg/kg DM |
| | W2 | 2.2 ± 0.3 | 2.9 ± 0.2 | |
| | W3 | 11.0 ± 0.5 | 7.3 ± 0.2 | |
| | W4 | 10.3 ± 1.5 | 8.9 ± 0.3 | |
| | W5 | 7.1 ± 0.2 | 9.2 ± 1.3 | |

## 4. Conclusions

Incubation of leachate in the presence of phosphate washing sludge and sugar lime sludge for 24 h allowed reducing successfully the bacterial load brought by the raw leachate and the removal of bad odors. The addition of the phosphate sludge with both 20% and 50% concentrations and the treatment of the 20% sugar lime sludge optimized significantly the moisture content of the windrows within the recommended range and increased the degradation of organic matter compared to the untreated control windrow. The pH, C/N ratio, microbiological quality, and humification parameters are within the standards of mature and stable composts. Notable results in terms of compost characteristics were recorded with phosphate sludge at 50% concentration compared to the 20% concentration. The treatment with phosphate sludge (20%) had a positive effect on compost parameters, hygienic quality, and humification process compared to the treatment of sugar lime sludge (20%). The use of phosphate washing sludge to valorize leachate from dump site by the production of compost seems to be a sustainable solution, which allows by the same way the valorization of phosphate washing sludge, which remains not valorized in many countries producing phosphates. Following these results, phytotoxicity tests, analyses of radioactivity in the different initial substrates used, as well as the final composts will be carried out to prove the quality of the composts from a toxicity point of view.

**Author Contributions:** M.M. roles/writing—original draft and formal analysis; B.I. and A.M. validation; K.F. supervision and writing—review and editing. All authors have read and agreed to the published version of the manuscript.

**Funding:** This research received no external funding.

**Institutional Review Board Statement:** Not applicable.

**Informed Consent Statement:** Not applicable.

**Data Availability Statement:** Not applicable.

**Conflicts of Interest:** The authors declare no conflict of interest.

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
