# Peer review of "The Use of Phosphate Washing Sludge to Recover by Composting the Leachate from the Controlled Landfill"

_processes, doi:10.3390/pr9101735_

Round 1
Reviewer 1 Report
The manuscript by Mobaligh and co-workers studies the possibility of composting leachate with green waste and phosphate washing sludge. Different conditions were tested and analyzed. According to the authors, the composts obtained with phosphate sludge were the most stable and mature and can be used in the agricultural field or green space.
Here some comments and corrections that should be addressed:
Line 13: The authors should provide an introduction to the abstract.
Line 80: Please provide the full meaning of OM and TOC.
Line 91: To enumerate the various origins of the waste, please use Roman numerals (e.g. (i)) or alphabet (e.g. (a)). Do not use topics with “•”.
Table 1: You must provide the full meaning of the abbreviations.
Lines 146 and 147 are unformatted.
Line 158 and 160: The scientific names should be in italic.
Line 183: There is something missing before: “[13] suggested…”
Line 187: Please delete the parentheses in this sentence: “is shown in (Figure 1).”
Line 190: You must provide the full words: doesn’t = does not
Line 202: Why is written 3.1. Subsection?
Please eliminate the boarder lines of the graphs.
Line 207: Please delete the parentheses in this sentence: “The results in (Figure 2) showed…”
Line 227: Please delete the parentheses in this sentence: “composting is shown in (Figure 3).”
Line 231: The 2 from CO2 should be subscripted.
Line 234: The sentence “From day 21st to day 35st” is incorrect (“35th").
Line 235: The 3+ from NH3+ should be subscripted.
Figure 5: May you please statistically compare the organic matter between different windrows?
Line 273: The scientific names should be in italic.
Figure 8: The different figures should be numerated and described in the captions. Figure 8 is not only about W5.
Lines 357 – 360: Please explain better this segment. Please explain what is composts C1, C2, C3, C4 and C5.
Line 377: Please delete the parentheses in this sentence: “As shown in (Figure 9)”
Line 385: There is something missing before: “[39] suggested that the form and…”
Line 400: Please correct: “color of the compost 4 be can less brown.”
Table 6: Could you provide supplementary figures representative of table 6?
Line 415: Please correct: “ Sex heavy metals: nickel, chromium, copper, zinc, cadmium, lead, arsenic…”
Line 423-424: Please correct SO42- and PO43-.
Table 7: The concentrations of heavy metals represented in table 7 correspond to Windrows but in the text is about composts.
Author Response
Dear Colleague,
Thank you for your comments. You can find bellow the answers to these comments.
Line 13: The authors should provide an introduction to the abstract.
Response: Done as requested (Page 1; Lines 13-15)
Line 80: Please provide the full meaning of OM and TOC.
Response: The full meaning of OM and TOC were provided as requested Page 3, under table 1.
Line 91: To enumerate the various origins of the waste, please use Roman numerals (e.g. (i)) or alphabet (e.g. (a)). Do not use topics with “•”.
Response: Done as requested (Page 3; Lines 101-109)
Table 1: You must provide the full meaning of the abbreviations.
Response: The full meaning of the abbreviations were provided as requested (Page 3; Lines 113-115)
Lines 146 and 147 are unformatted.
Response: The equation was formatted as requested (Page 5; Line 163)
Line 158 and 160: The scientific names should be in italic.
Response: The scientific names were put in italic as requested (Page 5; Line 174 and Line 176)
Line 183: There is something missing before: “[13] suggested…”
Response: The sentence was rectified to be more clear as requested (Page 6; Line 201)
Line 187: Please delete the parentheses in this sentence: “is shown in (Figure 1).”
Response: Done as requested (Page 6; Line 203)
Line 190: You must provide the full words: doesn’t = does not
Response: Done as requested (Page 6; Line 206)
Line 202: Why is written?
Response: this is a typing error. The term “3.1. Subsection” was deleted (Page 6; Line 219)
Please eliminate the boarder lines of the graphs.
Response: Done as requested in all the graphs
Line 207: Please delete the parentheses in this sentence: “The results in (Figure 2) showed…”
Response: Done as requested (Page 6; Line 224)
Line 227: Please delete the parentheses in this sentence: “composting is shown in (Figure 3).”
Response: Done as requested (Page 7; Line 244)
Line 231: The 2 from CO2 should be subscripted.
Response: Done as requested (Page 7; Line 248)
Line 234: The sentence “From day 21st to day 35st” is incorrect (“35th").
Response: The sentence was corrected as requested (Page 7; Line 251)
Line 235: The 3+ from NH3+ should be subscripted.
Response: Done as requested (Page 7; Line 252)
Figure 5: May you please statistically compare the organic matter between different windrows?
Response: Thank you very much for this pertinent comment. The answer to this question already exists in the main text (Page 9; Lines 277-283)
Line 273: The scientific names should be in italic.
Response: The scientific names were put in italic as requested (Page 9; Line 290)
Figure 8: The different figures should be numerated and described in the captions. Figure 8 is not only about W5.
Response: Done as requested (Page 13; Lines 368-369)
Lines 357 – 360: Please explain better this segment. Please explain what is composts C1, C2, C3, C4 and C5.
Response: Done as requested (Page 13; Lines 378)
Line 377: Please delete the parentheses in this sentence: “As shown in (Figure 9)”
Response: Done as requested (Page 14; Line 394)
Line 385: There is something missing before: “[39] suggested that the form and…”
Response: The sentence was rectified to be more clear as requested (Page 14; Line 402)
Line 400: Please correct: “color of the compost 4 be can less brown.”
Response: The sentence was rectified to be more clear as requested (Page 14; Line 417)
Table 6: Could you provide supplementary figures representative of table 6?
Response: The supplementary figure representative of table 6 is provided as requested (Supplementary material). Look at please to the attached file.
Line 415: Please correct: “ Sex heavy metals: nickel, chromium, copper, zinc, cadmium, lead, arsenic…”
Response: The sentence was rectified to be clear as requested (Page 15; Line 432).
Line 423-424: Please correct SO42- and PO43-.
Response: Done as requested (Page 15; Lines 440-441)
Table 7: The concentrations of heavy metals represented in table 7 correspond to Windrows but in the text is about composts.
Response: The sentence was rectified to be more clear as requested (Page 15; Line 443).
Yours sincerely,
khalid FARES

Reviewer 2 Report
The article deals with an interesting topic and markedly problematic residual substrates are managed through composting. However, the following suggestions should be considered before publication:
- The authors state in abstract and conclusions that the generation of bad odor is achieved by co-composting. However, the results derived from the measurement of such variable through a specific instrument (olfactometer) have not been included in the manuscript.
- INTRODUCTION: The authors should include the annual generation rate of all the wastes evaluated in their study. In addition, the objective of the article should be rephrased and its novelty highlighted.
- MATERIALS AND METHODS: All the acronysms should be defined (for example, those related to physicochemical analyses). Regarding the windrows, their dimensions and how often they were turned should be indicated, as well as the criterium to fix the amount of green waste added.
- RESULTS AND DISCUSSION: The authors should improve the discussion of the different subsections by comparing their findings with others already published by other authors. Line 248-249: please revise the statement. If total nitrogen is adequately measured, this should include the nitrogen in form of proteins. Section 3.8: How was color and texture measured objectively? Table 7: The sense of the last column of the table is not clear.
- CONCLUSIONS: The authors should improve the quality of the manuscript by including some results of the preliminary phytotoxicity tests they mention in the last part of this section.
Author Response
Dear Colleague,
Thank you for your comments. You can find bellow the anserws to thes comments.
The authors state in abstract and conclusions that the generation of bad odor is achieved by co-composting. However, the results derived from the measurement of such variable through a specific instrument (olfactometer) have not been included in the manuscript.
Response: Thank you very much for this pertinent comment. Indeed, the bad odors of leachate disappear after addition of phosphate washing sludge or sugar lime sludge. This result was so evident that we did not see any need to measure the odors by olfactometer for example.
INTRODUCTION: The authors should include the annual generation rate of all the wastes evaluated in their study. In addition, the objective of the article should be rephrased and its novelty highlighted.
Response: The annual generation rate of all the wastes evaluated in this study were already provided Page 1; Line 39 and Page 2; Line 54 and Line 59). We have added Page 2, Line 54 the quantity of phosphate washing sludge produced in Morocco.
The objective of the article was rephrased and its novelty highlighted to be clearer as requested (Page 2; Lines 83-90).
MATERIALS AND METHODS: All the acronysms should be defined (for example, those related to physicochemical analyses). Regarding the windrows, their dimensions and how often they were turned should be indicated, as well as the criterium to fix the amount of green waste added.
Response: All information were provided in MATERIALS AND METHODS section:
Acronyms: (Page 3; Lines 113-115, Page 4; Lines 123-124 and Line 143).
The windrows dimensions and how often they were turned : Page 4; Line 134 and line 145.
The criterium to fix the amount of green waste was added in the main text as requested (Page 4; Line 132).
RESULTS AND DISCUSSION: The authors should improve the discussion of the different subsections by comparing their findings with others already published by other authors.
Response: Thank you very much for this pertinent comment. The discussion was made based on the existing results in the literature ; however, very few results were close to our study. Its a new work never published before.
Line 248-249: please revise the statement. If total nitrogen is adequately measured, this should include the nitrogen in form of proteins.
Response: The statement was revised to be clear in the main text (Page 8; Line 266).
Section 3.8: How was color and texture measured objectively?
Response: Color was determined visually and texture was determined using different size sieves (200 µm to 2 mm)
Table 7: The sense of the last column of the table is not clear.
Response: The sense of the last column of the Table 7 was rectified to be more clear in the title (Page 15; Line 447).
CONCLUSIONS: The authors should improve the quality of the manuscript by including some results of the preliminary phytotoxicity tests they mention in the last part of this section.
Response: The last part of the Conclusions section was revised to be clear in the main text (Page 16, Line 466) : the phytotoxicity test was not carried out yet.
The results of this test as well as other results will be published in another paper.
Round 2
Reviewer 1 Report
Mobaligh and co-workers modified the manuscript according to the suggestions. Now I believe the paper is suitable for publishing.